# Astroglial Connexins Inactivation Increases Relapse of Depressive-like Phenotype after Antidepressant Withdrawal

**DOI:** 10.3390/ijms232113227

**Published:** 2022-10-30

**Authors:** Benjamin Portal, Flora Vasile, Jonathan Zapata, Camille Lejards, Abd El Kader Ait Tayeb, Romain Colle, Céline Verstuyft, Emmanuelle Corruble, Nathalie Rouach, Bruno P. Guiard

**Affiliations:** 1Centre de Recherches sur la Cognition Animale, Centre de Biologie Intégrative, Université de Toulouse, CNRS, 31062 Toulouse, France; 2Neuroglial Interactions in Cerebral Physiopathology, Center for Interdisciplinary Research in Biology, Collège de France, CNRS UMR 7241, INSERM U1050, Labex Memolife, PSL Research University, 75005 Paris, France; 3CESP, MOODS Team, INSERM, Faculté de Médecine, University of Paris-Saclay, 94275 Le Kremlin Bicêtre, France; 4Service Hospitalo-Universitaire de Psychiatrie de Bicêtre, Hôpitaux Universitaires Paris-Saclay, Assistance Publique-Hôpitaux de Paris, Hôpital de Bicêtre, 94275 Le Kremlin Bicêtre, France; 5Service de Génétique Moléculaire, Pharmacogénétique et Hormonologie de Bicêtre, Hôpitaux Universitaires Paris-Saclay, Assistance Publique-Hôpitaux de Paris, Hôpital de Bicêtre, 94275 Le Kremlin Bicêtre, France

**Keywords:** major depression, antidepressant, monoamines, astrocytes, connexins

## Abstract

Studies suggest that astrocytic connexins (Cx) have an important role in the regulation of high brain functions through their ability to establish fine-tuned communication with neurons within the tripartite synapse. In light of these properties, growing evidence suggests a role of Cx in psychiatric disorders such as major depression but also in the therapeutic activity of antidepressant drugs. However, the real impact of Cx on treatment response and the underlying neurobiological mechanisms remain yet to be clarified. On this ground, the present study was designed to evaluate the functional activity of Cx in a mouse model of depression based on chronic corticosterone exposure and to determine to which extent their pharmacological inactivation influences the antidepressant-like activity of venlafaxine (VENLA). On the one hand, our results indicate that depressed mice have impaired Cx-based gap-junction and hemichannel activities. On the other hand, while VENLA exerts robust antidepressant-like activity in depressed mice; this effect is abolished by the pharmacological inhibition of Cx with carbenoxolone (CBX). Interestingly, the combination of VENLA and CBX is also associated with a higher rate of relapse after treatment withdrawal. To our knowledge, this study is one of the first to develop a model of relapse, and our results reveal that Cx-mediated dynamic neuroglial interactions play a critical role in the efficacy of monoaminergic antidepressant drugs, thus providing new targets for the treatment of depression.

## 1. Introduction

Major depression (MD) is a common mental disorder affecting more than 350 million people worldwide. This pathology imposes a considerable burden on patients, and unfortunately, the underlying mechanisms remain partially understood. The monoaminergic hypothesis of MD has led to the development of antidepressant drugs enhancing serotonin (5-HT) and/or norepinephrine (NE) neurotransmission. Although effective, these treatments present several limitations, such as a long delay of action and a high rate of relapse [1]. This prompts research to identify new pharmacological targets at the cellular and molecular levels. Recent studies emphasize the importance of astrocytes in mental health, and the first type of evidence connecting these glial cells to MD relies on cell counting studies. For instance, postmortem immunohistochemical studies demonstrated a decrease in the number of GFAP and S100β positive cells in the prefrontal cortex [2] and the hippocampus [3] of depressed patients compared to age-matched nondepressed controls. Similar results were reported in animal models of depression, whereas antidepressant drugs reversed these impairments [4]. The second line of evidence linking glial cells and MD relates to altered astrocytic protein function, including glutamine synthetase, glutamate transporters, aquaporin-4 (AQP4) and connexins [5].

Connexins belong to a family of cellular transmembrane proteins, forming pores called connexons. In the brain, Cx30 and Cx43 are the main connexins (Cx) expressed by astrocytes [6]. If two connexons from neighboring cells bind, they form a gap junction (GJ), allowing communication between astrocytes thanks to the passive diffusion of small signaling molecules [7]. Alternatively, unopposed connexons compose hemichannels (HCs), ensuring exchanges between the cytoplasm and extracellular space [8]. In particular, HCs contribute to the release of gliotransmitters (i.e., glutamate, D-serine, ATP, GABA), key actors for synaptic plasticity and neurotransmission [5]. Clinical studies revealed reduced expression of Cx in the brain of depressed patients [9] and in relevant animal models [10]. The influence of antidepressant drugs on astrocytic Cx expression and GJ/HC activity has also been studied, unfortunately leading to controversies. Indeed, although in vivo and in vitro studies reported increased GJ activity in response to the antidepressants fluoxetine, fluvoxamine or venlafaxine [11,12], opposite results have been yielded with the same compounds using primary cultures of astrocytes [13]. Interestingly, all antidepressants tested so far showed an unambiguous inhibitory effect on HC activity [13]. The latter results strongly support the hypothesis that the blockade of HCs could improve the therapeutic activity of antidepressant drugs, but pharmacological tools specifically targeting this function are lacking. In a recent study, we provided evidence that the genetic inactivation of Cx43 in mice potentiates the acute antidepressant-like activity of fluoxetine in the tail suspension test in relation to greater serotonergic neurotransmission in the hippocampus [14]. Interestingly, the intrahippocampal injection of shRNA-CX43 in this brain region also has beneficial effects on fluoxetine response, thereby paving the way for the development of potentiating strategies based on the combination of antidepressant drugs and Cx43 inhibitors [14].

The present study explores the relationship between connexin functions and antidepressant response. To do so, we used a mouse model of depression based on chronic corticosterone exposure [15]. Using electrophysiological approaches, we sought to determine the excitability of astrocytes and the functional activity of GJs and HCs in a pathological context. We then used a behavioral approach to investigate the effects of carbenoxolone (CBX), a Cx blocker, on the efficacy of the serotonin/noradrenaline reuptake inhibitor, venlafaxine (VENLA). Going further, and based on the clinical definition, we tested for the first time the relapse in this mouse model. In particular, we raised the possibility that the pharmacological inactivation of Cx43 might influence this parameter after VENLA discontinuation.

## 2. Results

### 2.1. Chronic Corticosterone Depolarizes Astrocytes and Reduces Gap-Junction Coupling and Hemichannel Activity

We first investigated the effect of chronic CORT on the membrane properties of astrocytes in hippocampal slices. Patch-clamp recordings of astrocytes revealed that CORT depolarized the astrocyte’s membrane (*p* < 0.001), whereas capacitance and membrane resistance remained unchanged (*p* > 0.05 and *p* > 0.05, respectively; Figure 1A). We then assessed GJ-mediated intercellular communication of astrocytes by analyzing astroglial diffusion of biocytin, a GJ-permeable tracer dialyzed in a single astrocyte by patch-clamp (Figure 1B). We found a decreased diffusion of biocytin within the astroglial network in CORT mice compared to controls (*p* < 0.01), indicating reduced GJ coupling. Since Cx also forms HCs, which mediate direct exchanges with the extracellular space, we assessed their activity using the ethidium bromide (EtBr) uptake assay (Figure 1C–E). We found that HCs were activated in control mice, since EtBr uptake was reduced by carbenoxolone (CBX; 200 µM), a nonselective Cx pharmacological blocker, or by the Gap26 mimetic peptide (100 µM), a Cx43 blocker that preferentially targets HCs (*p* < 0.05 and *p* < 0.01, respectively; Figure 1C). In contrast, HCs were inhibited in CORT mice, as CBX or Gap26 had no effect on EtBr uptake by astrocytes (*p* > 0.05 and *p* > 0.05, respectively; Figure 1C). In addition, the Gap26 scramble peptide (Gap26sc) or the pharmacological pannexin blocker probenecid (PBN; 1 mM) [16] had no effect on HC activity of astrocytes in control (*p* > 0.05 and *p* > 0.05, respectively) and CORT mice (*p* > 0.05 and *p* > 0.05, respectively). This indicates that the astroglial HC activity found in control mice was selectively mediated by Cx43 HCs, but not by pannexin channels, and was inhibited by the chronic CORT treatment.

### 2.2. Pharmacological Alteration of Astrocytic Connexins Hinders Venlafaxine Response

We next carefully followed a group of mice chronically exposed to CORT (or its vehicle) by assessing first the state of their fur coat as a reliable and well-validated index of a depressive-like state [17]. A progressive deterioration was detected in CORT mice compared to control animals, with a significant higher score detected after 10 weeks of exposure (0.07 ± 0.005 (*n =* 12) vs. 0.36 ± 0.01 (*n* = 48); *p* < 0.001)) (Appendix A). Once this period was reached, mice still under CORT were separated into four groups, receiving either VEH, VENLA (15 mg/kg/day), CBX (10 mg/kg/day) or the combination of VENLA/CBX for 8 additional weeks. At Week 18, we deployed a comprehensive battery of behavioral tests to address the core symptoms of depression (Figure 2A). On the one hand, we observed that CORT and also VENLA and CBX did not alter locomotor activity (Figure 2B). On the other hand, we confirmed that CORT-exposed mice display neurobehavioral anomalies. These impairments included a high level of anxiety measured in the OF and NSF (Figure 2B,C) and higher resignation evaluated in the TST (Figure 2D) compared to nondepressed control VEH mice. Interestingly, self-care evaluated in the ST was unchanged between groups (Figure 2E). Based on mice behavior in the latter paradigms, an emotionality z-score was then calculated (Figure 3). One-way analysis on emotionality z-score at Week 18 (EM1; Figure 3A) revealed a significant effect of treatment factor (F(4,45) = 13.553, *p* < 0.001; Figure 2A). As expected and previously reported [10], post-hoc tests showed that chronic CORT significantly increased this score compared to control VEH animals (*p* < 0.001). Interestingly, the emotionality z-score was completely reversed in CORT/VENLA compared to CORT/VEH (*p* < 0.001) but not in CORT/CBX (*p* = 0.26) and CORT/VENLA/CBX mice (*p* = 0.87).

Because in clinical trials antidepressant treatment response is defined as a >50% improvement in the severity of depression [18,19], we used this index to determine the number of responders in each group receiving pharmacological compounds as previously used in preclinical studies [20]. In particular, we used the emotionality z-score of CORT mice as a reference for the pathological state (without treatment) and determined the number of individuals displaying a reduction of z-score of at least 50% of the baseline (i.e., CORT mice). Doing so, we found that CORT/VENLA mice displayed a higher rate of response compared to CORT/VENLA/CBX mice (80% vs. 8%; Fisher’s exact test; *p* < 0.001; Figure 3B, green dots).

To model relapse in the same mice, we stopped pharmacological treatments while maintaining mice under CORT. At Week 22, after 3 weeks of treatment withdrawal, a new emotionality z-score (EM2) was calculated (Figure 3C). A significant effect of treatment factor was evidenced (F(4,45) = 35.768, *p* < 0.001), and post-hoc analysis indicated that CORT/VEH (*p* < 0.001), CORT/VENLA (*p* < 0.001), CORT/CBX (*p* < 0.001) and CORT/VENLA/CBX (*p* < 0.001) animals displayed higher z-score than that calculated in control VEH mice. However, no differences between all the four groups exposed to CORT were unveiled, whatever the treatment received before their withdrawal. Considering the CORT/VEH group as the reference for depressed animals, the elevation of the emotionality z-score in CORT/VENLA, CORT/CBX and CORT/VENLA/CBX suggested that some animals experienced relapse. We then focused on individual performances after treatment withdrawal (Figure 3D). Assuming that the threshold of depressive phenotype can be defined by the averaged post-withdrawal score measured in the CORT/VEH mice, we evaluated the effect of VENLA, CBX or their combination on the relapse rate. Statistical analysis revealed a significant effect of treatment factor (F(2,20) = 5.857, *p* = 0.01). We found that 10% of the CORT/VENLA mice, 60% of the CORT/CBX mice (vs. CORT/VENLA; Fisher’s exact test; *p* = 0.057) and 75% of the CORT/VENLA/CBX mice (vs. CORT/VENLA; Fisher’s exact test; *p* = 0.004) reached this threshold 3 weeks after treatment withdrawal.

## 3. Discussion

In this study, we modeled a depressive-like phenotype in mice using chronic corticosterone (CORT) exposure. Our results indicate that mice chronically injected with the stress hormone in the drinking water display impairments of Cx-based gap-junction (GJ) and hemichannel (HC) activities, along with behavioral anomalies recapitulating core symptoms of depression (Figure 4). Indeed, using an emotionality z-score, which has proven its relevance to studying depression as a syndrome and refining its translational applicability [21], our results indicated that CORT induced a robust and long-lasting depressive-like phenotype as previously demonstrated [15]. Interestingly, the antidepressant venlafaxine (VENLA) reversed the effect of CORT, while it failed to do so in the presence of the Cx blocker carbenoxolone (CBX). The negative impact of CBX on treatment response was also characterized by a higher rate of relapse in mice receiving the combination of VENLA/CBX compared to VENLA alone.

Our findings regarding GJ activity are consistent with previous in vivo studies, showing that both the expression and the function of these molecular elements are blunted in different animal models of depression [11,22,23]. Although the reason for such functional modifications is not elucidated, one can wonder if the ability of CORT to depolarize astrocytes is not responsible for these functional changes. In the current state of knowledge, experimental data suggest that high K+ concentration-induced astrocytic depolarization is associated with an increase, rather than a decrease, in GJ activity [24,25]. Regarding HC activity, our findings stand in contrast with data showing that chronic restraint stress opens HC to promote the release of hippocampal glutamate in neurotoxic concentrations [26]. It is thus possible that the nature of the stress (physical vs. pharmacological) differentially affects astrocytic Cx (i.e, GJ vs. HC activity). Alternatively, the present data raise the possibility that the impairment of GJ is sufficient to induce a depressive-like phenotype regardless of the functional status of the HC. In support of the latter hypothesis, we observed that the nonselective Cx blocker CBX, which inhibits both GJs and HCs, dampened VENLA-induced antidepressant-like effects. The inactivation of astroglial Cx functions thus has a negative impact on emotionality in response to venlafaxine treatment and more specifically on its ability to restore a normal phenotype. Whether or not such a property can be extended to other antidepressant treatments remains to be determined. It would also be interesting to demonstrate that the inhibitory effect of CBX on Cx is not modified in VENLA-treated animals. Interestingly, the literature regarding the link between Cx and antidepressant drugs is still a matter of debate. On the one hand, it has been shown that antidepressant drugs promote upregulation of GJ [11,12,13]. On the other hand, an unambiguous inhibitory action of SSRIs but also of norepinephrine reuptake inhibitors and serotonin-norepinephrine reuptake inhibitors on HC functional activity has been reported [13]. Moreover, evidence suggests that the genetic or pharmacological inactivation of Cx43 potentiates the response of antidepressant drugs at the behavioral, neurochemical and electrophysiological levels [14,27]. Consequently, these findings led us to envision that antidepressant drugs may favor the opening of GJs [11] but compromise HC function [13]. This aspect warrants further investigations using more specific pharmacological tools able to discriminate GJs and HCs. A recent study shows that the specific inactivation of astrocytic HCs by INI-0602 attenuated depressive-like behaviors [28], thereby reinforcing the idea that GJs and HCs are differentially regulated by antidepressant drugs and, reciprocally, that antidepressant drugs differentially influence GJs and HCs. Beyond these considerations, it is noteworthy that the effects of CBX herein reported could result from its interactions with other yet undescribed targets, thus exerting direct or indirect effects upon emotional states. For example, it has been reported that CBX inhibits 11-β-hydroxysteroid dehydrogenase [29], therefore competing with the synthesis of endogenous corticosterone. However, in our experimental conditions and in agreement with previous findings [30], we were not able to unveil changes in plasma corticosterone levels in response to CBX (Appendix A). Beyond these considerations, it is now well accepted that the heart and the brain have tight anatomical and functional interaction connections. In particular, through the neuroendocrine–heart axis and the neuroimmune–heart axis, the brain maintains physiological homeostasis of the cardiovascular system, which in turn controls brain blood flow [31]. Since the heart is endowed with a high density of Cx43 controlling conduction velocity [32], one could expect that the systemic administration of CBX may also inhibit Cx43 in the heart, causing low cardiac output, brain hypoperfusion and subsequent abnormal functions, such as the development of depressive behavior [33]. The latter hypothesis is supported by epidemiological data showing that patients with cardiovascular diseases are more likely to be depressed [31] and possibly display poorer treatment response, as reported here.

Importantly, our study also focused on relapse, a poorly considered parameter in preclinical studies. In patients, relapse is mostly due to a constant high level of stress and related hormones, notably after treatment withdrawal [34]. By anthropomorphic considerations, we thus modeled relapse by stopping pharmacological treatments while maintaining mice under CORT for 3 weeks. We observed that the emotionality z-scores in VENLA, CBX and VENLA/CBX groups returned to the level of CORT mice. This recovery of pathological state differed between groups and a higher rate of relapse was detected in mice receiving the VENLA/CBX compared to VENLA alone. Thus, mice treated with the combination of VENLA/CBX could be more prone to relapse than mice treated with VENLA alone.

The neurobiological mechanisms underlying the relationship between astroglial connexins and antidepressant efficacy have been poorly investigated. It is well known that the antidepressant-like effects of VENLA at the dose used in the present study rely on its ability to increase extracellular serotonin and norepinephrine levels [35]. Consequently, it is possible that the uncoupling of neuroglial communication in CORT mice leads to alterations of neurotransmission that can be detrimental to emotional behaviors and treatment response. Further experiments are thus needed to determine whether CBX impairs VENLA-induced enhancement of synaptic monoaminergic tone in the hippocampus or other brain areas involved in the regulation of mood. The latter aspect is important since evidence demonstrates that monoaminergic antidepressant drugs stimulate the release of small signaling molecules, notably through Cx43 [36]. Hence, it is possible that CBX limits the accumulation of gliotransmitters displaying antidepressant-like effects as ATP [37] or D-serine [38] in the synaptic cleft. On this ground, one could speculate that CBX reinforces the negative influence of CORT on both GJs and HCs, as reported in this study. Moreover, given that monoaminergic antidepressants including VENLA stimulate adult hippocampal neurogenesis [39], we can anticipate that the inactivation of Cx negatively reverberates in such processes [40]. It is interesting to note that a reduction in Cx43 expression attenuates myelination [41]. Since chronic treatment with VENLA has been described to counteract demyelination [42], one would expect that astrocytic connexin inactivation dampened the ability of antidepressants to enhance oligodendrocyte-induced myelination.

## 4. Materials and Methods

### 4.1. Animals

Seven-week-old adult C57Bl/6J male mice were purchased from Janvier Laboratories (Le Genest-Saint-Isle, France) and housed by two to four per cage, with a 12/12 h day/light cycle (light on at 8:00 a.m). Food and water were available ad libitum. All experimental procedures were conducted in accordance with the European Directive 2010/63/EU and were approved by the French Ministry of Research and the Local Ethics Committee (APFIS No. 2018100110245946#16913).

### 4.2. Drugs

Corticosterone (CORT, Sigma-Aldrich, Illkirch, France) was dissolved in vehicle solution (VEH, 0.45% β-cyclodextrin) and delivered in drinking water (5 mg/kg) for 23 weeks in opaque bottles. Carbenoxolone (CBX, Sigma-Aldrich, Illkirch, France) was dissolved in 0.9% NaCl solution and delivered daily via intraperitoneal injections (10 mg/kg/day) for 9 weeks. Venlafaxine (VENLA, Sigma-Aldrich, Illkirch, France) was dissolved in CORT solution and delivered in drinking water (15 mg/kg/day) for the same duration.

### 4.3. Electrophysiology

Acute transverse hippocampal slices (400 µm) were prepared as previously described [43]. Slices were maintained at room temperature in a storage chamber that was perfused with an artificial cerebrospinal fluid (aCSF). Slices were then transferred to a submerged chamber mounted on an Olympus BX51WI microscope and were perfused with aCSF at a rate of 1.5 mL/min at room temperature. Somatic whole-cell recordings were obtained from visually identified astrocytes, using 5–8 MΩ glass pipettes filled with (in mM) 105 k-gluconate, 30 KCl, 10 HEPES, 10 phosphocreatine, 4 ATP-Mg, 0.3 GTP-Tris and 0.3 EGTA (pH 7.4, 280 Mohm l21). Recordings were acquired with Axopatch-1D amplifiers (Molecular Devices, San Jose, California, United States of America) digitized at 10 kHz, filtered at 2 kHz, stored and analyzed on a computer using CLAMPFIT software (Molecular Devices, San Jose, California, United States of America). Intercellular coupling was assessed thanks to the diffusion of biocytin (2 mg/mL), which diffused passively in astrocytes for 20 min in current-clamp mode.

### 4.4. Dye Uptake by Hemichannels

To investigate the contribution of Cx43 hemichannels, slices were incubated 15 min before and during the application of ethidium bromide (EtBr; 314 Da, 4 μM), a hemichannel-permeable fluorescent tracer, with CBX (200 µM), Gap26 (100 µM) or Gap26 scramble peptide (100 µM). Slices were then rinsed for 15 min in aCSF, fixed for 2 h in 4% paraformaldehyde, immunostained for GFAP (rabbit anti-GFAP, Sigma-Aldrich, 1:500), with anti-rabbit antibodies conjugated to Alexa Fluor 647 (1:1000, Life Technologies), and mounted on fluoromount. Labeled cells were examined with a confocal laser-scanning microscope (TCS SP5, Leica). Stacks of consecutive confocal images with high-bit depth color (16 bit) were taken at 1 µm intervals and acquired in sequential mode with lasers (647 nm for GFAP and 555 nm for EtBr). Dye uptake analysis was performed in stratum radiatum CA1 astrocytes that were positive for GFAP. Dye uptake was evaluated and expressed as the difference between the fluorescence measured in astrocytes (20–30 per slice) and the background fluorescence measured in the same field where no labeled cells were detected.

### 4.5. Behavioral Analysis

We performed a longitudinal study in which each animal was tested in two sets of behavioral paradigms, including the fur coat state, the open field test (OF), the novelty-suppressed feeding (NSF), the tail suspension test (TST) and the splash test (ST).

#### 4.5.1. Coat State

The fur coat state refers to another symptom of depression: the lack or reduction of self-care. The fur state was assessed across eight areas of the body (i.e., head, neck, body, paws tail). For one area of interest, a score of 0 was attributed if the fur was normal. On the contrary, a score of 0.5 or 1 was attributed if the fur was greasy and/or degraded. The mean score calculated over the eight areas of interest was used to compare the fur coat state between animals; the more the animal is stressed, the higher the score.

#### 4.5.2. Open Field Test (OF)

This test is based on the natural tendency of rodents to avoid open spaces considered as anxiogenic [44]. Animals were free to explore a spatial-cue-free circular arena (diameter 40 cm) for 10 min. Virtually divided into two areas (central area and peripheral area), the time spent in the central area, as well as the number of entrances in that zone, was automatically recorded thanks to Ethovision software (Noldus tech., The Netherlands). The more the animal avoided the central area, the more it was considered anxious and vice versa.

#### 4.5.3. Novelty-Suppressed Feeding Test (NSF)

Based on the conflict between hunger and fear of open spaces [15], the NSF test was used to assess anxiety. After a 24 h starving period, animals were placed in an open field (40 cm × 60 cm) containing in the center a pellet of food brightly enlightened. The time spent in the area containing the pellet of food was used as an indicator of the anxiety state of the animals. A cut-off to stop the experiment was placed at 10 min. Food consumption in the home cage was immediately assessed for 5 min after the test, to ensure that there was no difference in hunger levels between each animal.

#### 4.5.4. Tail Suspension Test (TST)

The TST measures resignation, one of the symptoms of depression [45]. Mice are hanged by the tail to a hook placed at the top of an opaque box (dimensions: 16 cm × 38 cm × 15 cm). The total time of immobility is recorded over a 6 min trial. The greater the immobility, the more resigned the mouse was considered to be and vice versa

#### 4.5.5. Splash Test (ST)

ST refers to another symptom of depression, namely, self-care. This test consisted in squirting 200 µL of a 10% sucrose solution on the mouse’s snout. The grooming frequency was then recorded for 5 min. This test was realized in the home cage in order to avoid exploration-related biases [15]. The lower the grooming, the more depressed the mouse was considered to be and vice versa.

An emotionality z-score was then used to integrate mice behavioral performances in several tests and evaluate the severity of depressive-like symptoms, as well as the number of animals that experienced a treatment response or a relapse in each experimental group [20]. By anthropomorphic analogy, we consider an animal as a “responder” when its emotionality z-score improved by at least 50% relative to the average of the CORT group. Regarding relapse, this parameter was defined as a post-drug withdrawal emotionality z-score, equal or above the average z-score of the CORT group.

### 4.6. Statistical Analysis

Regarding astrocytic electrophysiological parameters and intracellular coupling, Student’s *t*-tests were used to compare control VEH and CORT mice. A Kruskal–Wallis, post-hoc Dunn’s multiple comparisons test was applied to evaluate the effect of the various blockers on HC activity. For behavioral experiments, one-way analysis of variance (ANOVA) with treatment as the main factor was applied, followed by a Bonferroni post-hoc test to compare the effects of treatments between groups. IBM SPSS software (v23, IBM Corp, Bois Colombes, France) was used to conduct these statistical analyses.

## 5. Conclusions

Overall, our findings reveal that astrocytic Cx-mediated dynamic interactions with neurons are controlled in a fine-tuned manner by environmental factors and that it plays a critical role in the efficacy of monoaminergic antidepressant drugs. This should be carefully considered to develop new antidepressant strategies based on the manipulation of GJs and HCs. In general, pharmacological modulators present poor specificity towards connexin isoforms and functions of either HC or GJ. This instigated the development of several peptides acting on Cx. Among the most often used are Gap26/Gap27, which reproduced parts of the amino acid sequence of Cx43. However, although it is well accepted that this peptide inhibits Cx43 [46], to our knowledge, no data in the literature consider this pharmacological compound as a Cx43-specific inhibitor, and its specificity towards HC has been challenged. The development of more selective tools should help to better decipher the link between these molecular elements and antidepressant drug response. In this prospect, the recent development of antibodies that inhibit Cx43 HC, but not GJ [47], represents a new hope.

## Figures and Tables

**Figure 1 ijms-23-13227-f001:**
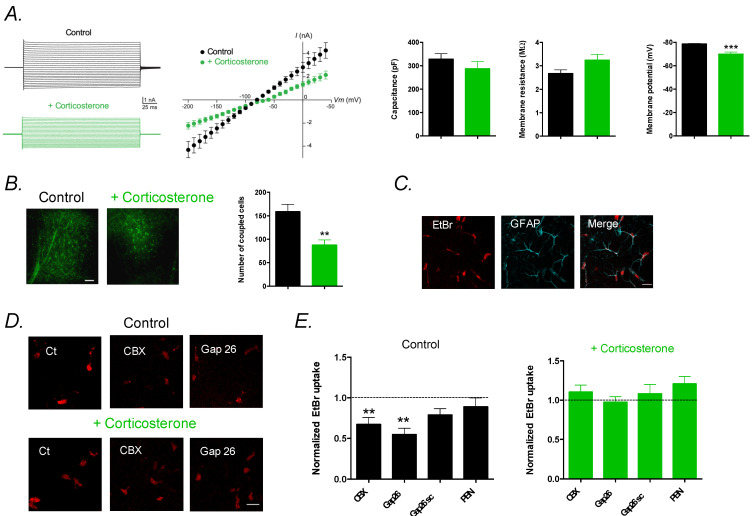
Chronic corticosterone treatment alters astrocytes’ membrane properties and astroglial connexin functions in the hippocampus. (**A**) Electrical properties of astrocytes with representative current–voltage (I–V) plots. Data are mean ± SEM of I–V curves, capacitance, membrane resistance and intrinsic membrane properties of astrocytes from control VEH mice (*n* = 7 cells from 5 mice) and corticosterone (CORT)-exposed mice (*n* = 8 cells from 5 mice). (**B**) Representative images and quantification of GJ-mediated biocytin coupling in CA1 stratum radiatum astrocytes from nontreated mice (*n* = 9 cells from 5 mice) and corticosterone-treated mice (*n* = 10 cells from 5 mice). Scale bar, 100 µm. Data are mean ± SEM of number of coupled cells. ** *p* < 0.01, and *** *p* < 0.001: significantly different from control VEH mice. (**C**) Representative basal EtBr uptake (red) in stratum radiatum astrocytes labeled with GFAP (cyan) in hippocampal slices. Scale bar, 20 µm. (**D**) Higher magnification of astroglial EtBr uptake (red) in hippocampal slices from control mice and mice treated with corticosterone, in basal conditions (Ct) and in the presence of CBX (200 µM), or Gap26 (100 µM), applied 15 min prior and during EtBr uptake assay. Scale bar, 20 µm. (**E**) Astrocytic EtBr uptake normalized to control conditions in slices from untreated mice (Control) incubated with CBX (200 µM, *n* = 13 slices from 5 mice), Gap26 (100 µM, *n* = 12 slices from 5 mice), Gap26 scramble peptide (Gap26 scr, 100 µM, *n* = 13 slices from 5 mice) or Probenecid (PBN, 1 mM, *n* = 8 slices from 3 mice) and from mice treated with corticosterone incubated with CBX (200 µM, *n* = 11 slices from 5 mice, Gap26 (100 µM, *n* = 13 slices from 5 mice, Gap26 scramble peptide (Gap26 scr, 100 µM, *n* = 12 slices from 5 mice, or Probenecid (PBN, 1 mM, *n* = 8 slices from 3 mice). Asterisks indicate statistical significance on raw data. ** *p*< 0.01: significantly different from the basal condition (no treatment) in control mice.

**Figure 2 ijms-23-13227-f002:**
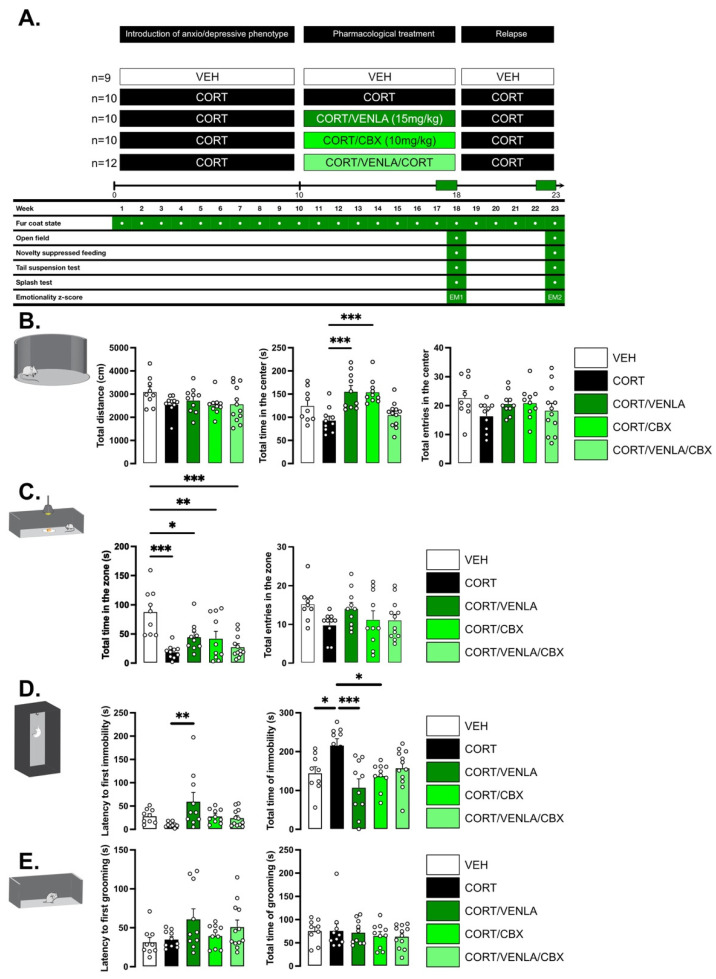
Anxio-/depressive-like behavior to study venlafaxine/carbenoxolone antidepressant effect in a mouse model of depression. (**A**) Design of experiment. Mice were chronically treated with VEH (*n* = 9 animals) or to CORT (*n* = 42) for 10 weeks. On Week 10, CORT group have been divided in four groups receiving either CORT (*n* = 10), CORT + VENLA (15 mg/kg/day in the drinking water; *n* = 10), CORT + CBX (10 mg/kg/day; i.p.; *n* = 10) or the combination of the three of them CORT + VENLA + CBX (*n* = 12). On Week 18, VENLA and CBX treatments were stopped, whereas CORT was maintained for all the groups. Emotionality z-score were calculated at Week 18 (EM1) and Week 23 (EM2) to respectively assess responsiveness and relapse to VENLA or VENLA + CBX treatment. (**B**–**E**) Anxio-/depressive-like behavior. Data are mean ± SEM of mice performances in the open field (**B**), novelty-suppressed feeding (**C**), tail suspension tests (**D**) and splash test (**E**). Asterisks indicate statistical significance on raw data. * *p* < 0.05, ** *p* < 0.01 and *** *p* < 0.001: Statistically different as indicated.

**Figure 3 ijms-23-13227-f003:**
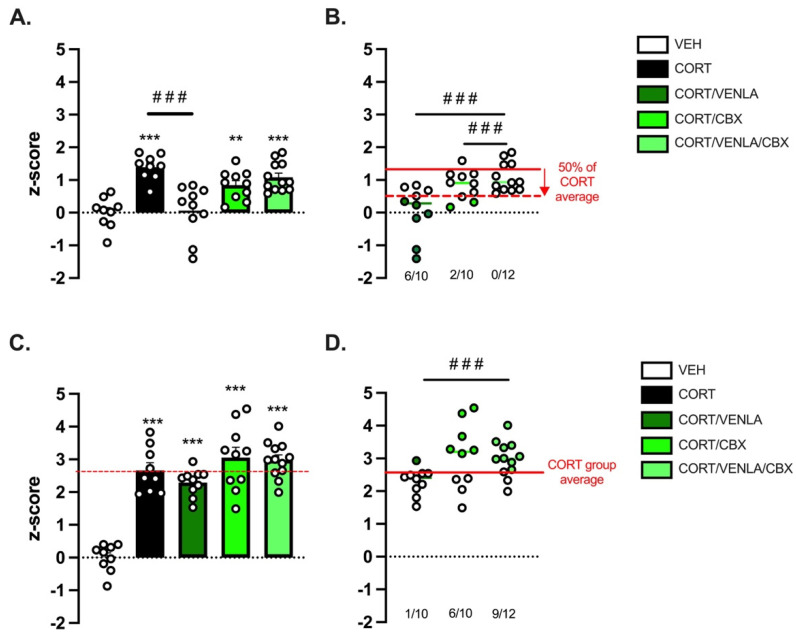
Inactivation of astroglial connexin functions hinders venlafaxine-induced antidepressant-like effects in the corticosterone mouse model of depression. Emotionality z-score was calculated based on a comprehensive battery of behavioral test ran at Week 18 (**A**,**B**) and Week 23 (**C**,**D**). A z-score >0 indicates a depressive-like state, whereas a score <0 indicates an antidepressant-like state. On Week 18 (**A**), an individual was considered as “responder to VENLA” when its z-score was <50% of the mean value of the CORT group (**B**, green dots). On Week 23 (**C**), after VENLA withdrawal, an individual was considered as “experiencing relapse” if its score was higher than the mean value of the CORT group (**D**, green dots). Asterisks indicate statistical significance on raw data: ** *p* < 0.01 and *** *p* < 0.005: statistically different from VEH group. Fisher’s exact test: ### *p* < 0.001: statistically different as indicated.

**Figure 4 ijms-23-13227-f004:**
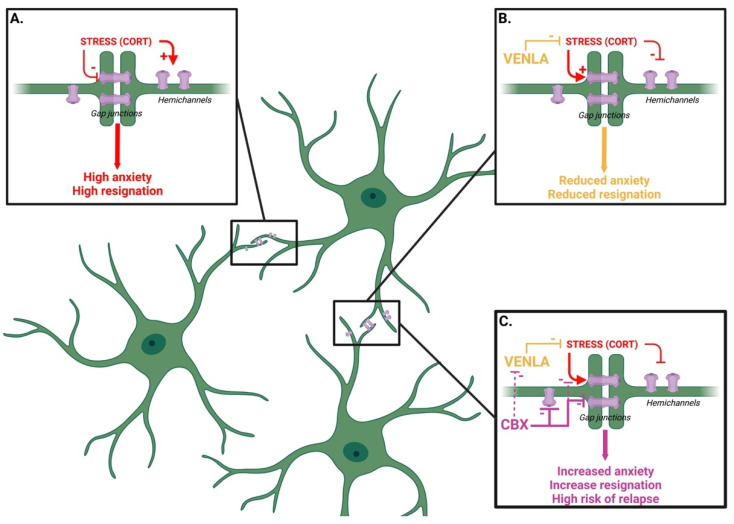
Schematic representation of the potential link between the astroglial connexins and the response to venlafaxine. In line with the literature, chronic stress (modeled here by the long-term exposure to corticosterone, CORT) is believed to block astroglial gap junctions (GJs) while maintaining hemichannels (HCs) opened. This configuration generates a high level of anxiety and resignation (**A**) suggesting that the impairment of GJ is sufficient to induce a depressive-like phenotype regardless of the functional status of the HCs. On the contrary, the long-term treatment with the antidepressant venlafaxine, either in direct opposition to CORT or by direct modulation of Cx, increases GJ and decreases HC expression/function. This configuration is associated with a reduced level of anxiety and resignation (**B**). The addition of the Cx blocker carbenoxolone (CBX) attenuates the response to venlafaxine and increases the risk of relapse after treatment withdrawal. Although CBX is not selective to GJs or HCs, this effect could result from direct inhibitory action on GJs (purple arrow). An interaction between CBX and venlafaxine (or CORT) catabolism (hatched purple arrow) is not excluded (**C**). Figure created with BioRender.com.

## Data Availability

Not applicable.

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
