# Peer review of "Astroglial Connexins Inactivation Increases Relapse of Depressive-like Phenotype after Antidepressant Withdrawal"

_ijms, 2022, doi:10.3390/ijms232113227_

Round 1

Reviewer 1 Report

The manuscript by Portal and colleagues presents interesting new results related to the effects of the connexins inhibitor carbenoxolone (CBX) on the antidepressant-like activity of venlafaxine in a mouse model of depression induced by chronically elevated corticosterone levels.

1. The authors show that the chronic treatment with CBX led to the abolishing of the effects of venlafaxine and an increase in relapse of depression-like behavior. However, CBX is known to be a very non-selective agent. The authors did not provide evidence that the effects of venlafaxine were abolished due to connexins inactivation, as stated in the conclusions and title of the manuscript.

The main conclusions of the manuscript are that “pharmacological inhibition of connexins abolishes antidepressant-like activity of venlafaxine in depressed mice and increases relapse of depression-like phenotype after venlafaxine withdrawal” (lines 28-30) and that “our results reveal that Cx-mediated dynamic neuroglial interactions plays a critical role in the efficacy of monoaminergic antidepressant drugs” (lines 31-33).

In the Discussion, the authors correctly point out that “it is noteworthy that the effects of CBX herein reported could result from its interactions with other yet undescribed targets, thus exerting direct or indirect effects upon emotional states. For example, it has been reported that CBX inhibits the 11-β-hydroxysteroid dehydrogenase [28], therefore competing with the synthesis of endogenous corticosterone.” (lines 247-251). This consideration also supports the lack of validity of the main conclusions of the manuscript. CBX exhibits a range of known activities that can affect emotional status. In addition to blocking connexin and pannexin channels, CBX directly affects GABA receptors, blocks NMDA- and P2X7-receptors, and Ca2+-channels (Connors, 2012, PMC3316363; Verselis&Srinivas, 2013, PMC3775990).

It is possible that CBX abolishes the effects of venlafaxine by acting through other known or unknown targets and having no additional effect on connexins in depressed mice treated with venlafaxine. There is no evidence in the manuscript against this possibility. Moreover, according to the data in the manuscript, connexins were already significantly inactivated in depressed mice compared to controls even in the absence of CBX, and CBX was not able to reduce hemichannel activity in astrocytes from depressed mice. Because there is no evidence in the manuscript that the treatment with CBX caused inhibition of connexins in venlafaxine-treated mice, it is very difficult to conclude that pharmacological inhibition of connexins, rather than CBX actions on any other targets, abolishes venlafaxine antidepressant activity and increases relapse of depressant-like behavior after antidepressant withdrawal.

Therefore, it is necessary either to show in the manuscript that CBX does inactivate connexins in mice treated with venlafaxine to prove the conclusions, or to reformulate the conclusions and discuss possible mechanisms of the reported effects of CBX.

Minor suggestions:

2.     It would be useful to indicate in the abstract the name of the Cx inhibitor used (carbenoxolone).

3.     Figure 3D: The designations of significance levels and group comparisons do not seem to correspond to the text in the Results (Lines 189, 190).

4.     Line 230: The inactivation of astroglial Cx functions has thus a negative impact on emotionality…. (it should be clarified: for example, “in venlafaxine-treated depressed mice”)

Typos:

5.     In the legends of Figs. 2 and 3, VELA was listed instead of VENLA.

6.     Line 167: VNELA.

7.     Decimal dots and commas are mixed up in the text (for example, line 201).

8.     Line 234: In the one hand

9.     Line 164: Mice we chronically treated with VEH (n=9 animals) or to CORT

Reviewer 2 Report

The manuscript entitled “Astroglial connexins inactivation increases relapse of depressive-like phenotype after antidepressant withdrawal” by Portal et al. suggests that astrocytic connexins play an important role in depression. The conclusions are mainly drawn from the observed effects of carbenoxolone in behavioral tests on mice exposed to chronic corticosterone treatment. In addition, the involvement of astrocytic connexins is also supported by in vitro measurements of connexin activity in response to corticosterone. Although the experimental results are generally in line with the conclusions, the uncertainty due to the non-specific action of the applied drugs and the alternative explanations of the experimental data should be discussed in much more detail.

The authors convincingly demonstrate in a multitude of anti-depressant behavioral tests that carbenoxolone (CBX) interferes with venlafaxine treatment. However, CBX is a well-known “dirty” drug that acts on various targets including VRAC channels, voltage-gated Ca2+ channels and many more. In addition, systemic administration of CBX may also inhibit connexins in the heart, causing reduction in heart rate and subsequent enervation of the animals that may be read out as depressive behavior. Most importantly, CBX is a known inhibitor of 11β-hydroxysteroid dehydrogenase, therefore it directly interferes with the corticosteroid metabolism. I acknowledge that evaluation of a more specific connexin inhibitor (e.g. a Cx43 antibody) would not be feasible in the timeframe of the revision of this manuscript. However, the authors should discuss in much more detail (apart from the few lines at the bottom of page 7) why do they think that the observed CBX effects should be attributed to connexin inhibition instead of a more apparent corticosteroid interference or other targets.

Minor issues:

Lines 99-109: The results on the effect of CBX and Gap26 are referenced to Figure 1C. However, Figure 1C shows only the EtBr uptake under control condition. The CBX and Gap26 data are presented in Figure 1D and E. In contrast, the real data of Figure 1C is not mentioned in the text.

Line 101, 108: Although Gap26 is a mimetic peptide based on the Cx43 sequence, no data exists in the literature that would demonstrate its subtype-specificity. Therefore, Gap26 should not be considered as a Cx43-specific inhibitor.

Figure 2B-E: VENLA is mistyped to VELA in all figures.

Reviewer 3 Report

Interesting article. The tasks set in the work are solved using adequate methods.

1) It is necessary to indicate the concentration of probenecid not only in the legend, but also in the text of the article. It also applies to other blockers. References to other works demonstrating that this blocker acts on panexins at the presented concentration will also be appropriate. This blocker is not very selective and has many other targets. If possible, I would recommend using 10Panx.

2) The quality and resolution of Figure 2 needs to be improved

3) A figure of the molecular mechanism resulting from the data obtained would be appropriate.

4) In the materials and methods, behavioral tests should be described in more detail, despite the presence of reference to previous work.

5) The discussion should discuss the role of Ca2+ ions, ATP release, etc. in the effects of Cx43 activation.

6) I did not see the conclusion to the article

Round 2

Reviewer 1 Report

The manuscript has been significantly improved in general agreement with the recommendations. I think the paper warrants publication.

Reviewer 3 Report

The article has been significantly improved. It may make sense for the authors to discuss the mechanisms of functioning of connexins. https://pubmed.ncbi.nlm.nih.gov/34439975/ https://pubmed.ncbi.nlm.nih.gov/34884629/ Optional